# Integration of eQTL and GEO Datasets to Identify Genes Associated with Breast Ductal Carcinoma In Situ

**DOI:** 10.3390/cimb47090747

**Published:** 2025-09-11

**Authors:** Cai-Qin Mo, Rui-Wang Xie, Wei-Wei Li, Min-Jie Zhong, Yu-Yang Li, Jun-Yu Lin, Juan-Si Zhang, Sheng-Kai Zheng, Wei Lin, Ling-Jun Kong, Sun-Wang Xu, Xiang-Jin Chen

**Affiliations:** 1Department of Thyroid and Breast Surgery, The First Affiliated Hospital of Fujian Medical University, Fuzhou 350005, China; mocaiqin1990@fjmu.edu.cn (C.-Q.M.); xierw1996@163.com (R.-W.X.); iewiew_leah@163.com (W.-W.L.); fjykdxzmj@126.com (M.-J.Z.); linjy2019@fjmu.edu.cn (J.-Y.L.); zhangjunsi2023@163.com (J.-S.Z.); zshengki@163.com (S.-K.Z.); lw@fjmu.edu.cn (W.L.); h23o17n21@163.com (L.-J.K.); 2Department of Thyroid and Breast Surgery, National Regional Medical Center, Binhai Campus of the First Affiliated Hospital, Fujian Medical University, Fuzhou 350212, China; 3Department of Pathology, The First Affiliated Hospital of Fujian Medical University, Fuzhou 350005, China; lyyang1218@126.com; 4Department of Pathology, National Regional Medical Center, Binhai Campus of the First Affiliated Hospital, Fujian Medical University, Fuzhou 350212, China; 5Fujian Provincial Key Laboratory of Precision Medicine for Cancer, Fuzhou 350005, China

**Keywords:** ductal carcinoma in situ, differentially expressed genes, Mendelian randomization, tumor microenvironment, gene function prediction

## Abstract

Background: Breast ductal carcinoma in situ (DCIS), a common precursor of breast cancer, has poorly understood susceptible driver genes. This study aimed to identify genes influencing DCIS progression by integrating Mendelian randomization (MR) and Gene Expression Omnibus (GEO) datasets. Methods: The GEO database was searched for DCIS-related datasets to extract differentially expressed genes (DEGs). MR was employed to find exposure single-nucleotide polymorphisms (SNPs) of expression quantitative trait locus (eQTL) gene expression from Genome-Wide Association Study database (GWAS) (IEU openGWAS project). DCIS was designated as the outcome variable. The intersection of genes was used for GO, KEGG and CIBERSORT analyses. The functional validation of selected DEGs was performed using Transwell invasion assays. Results: Four datasets (GSE7782, GSE16873, GSE21422, and GSE59246) and 19,943 eQTL exposure data were obtained from GEO and the IEU openGWAS project, respectively. By intersecting DEGs, 13 genes (LGALS8, PTPN12, YTHDC2, RNGTT, CYB5R2, KLHDC4, APOBEC3G, GPX3, RASA3, TSPAN4, MAPKAPK3, ZFP37, and RAB3IL1) were incorporated into subsequent KEGG and GO analyses. Functional assays confirmed that silencing PTPN12, YTHDC2 and MAPKAPK3, or overexpressing GPX3, RASA3 and TSPAN4, significantly suppressed DCIS cell invasion. These DEGs were linked to immune functions, such as antigen processing and presentation and the tumor microenvironment (TME), and they showed associations with dendritic cell activation differences. Conclusions: Thirteen genes were associated with DCIS progression, and six genes were validated in the cell experiments. KEGG and GO analyses highlight TME’s role in early breast cancer, enhancing understanding of DCIS occurrence and aiding identification of high-risk tumors.

## 1. Introduction

Breast ductal carcinoma in situ (DCIS), a precursor of invasive ductal cancer (IDC), is becoming more prevalent due to advancements in breast ultrasound and mammography. Approximately one in every four breast cancer patients is diagnosed with DCIS [1]. Without timely treatment, about 20–50% of DCIS patients may progress to IDC [2,3]. However, the molecular mechanisms underlying disease progression have remained unknown. Although the expression levels of CPA1, CUL7, LRRTM2, and POU2AF1 genes increase from normal tissue to DCIS and then to IDC [4], this information is insufficient to fully elucidate the natural progression of DCIS.

Gene Expression Omnibus (GEO) has been valuable in exploring tumor progression. Nevertheless, GEO-related data cannot directly reveal causal relationships between exposure and outcome factors. Therefore, even after identifying differentially expressed genes, the causal relationship between these genetic differences and the disease remains uncertain.

Mendelian randomization (MR) is a powerful approach that uses genetic variants (single nucleotide polymorphisms [SNPs]) as instrumental variables (IVs) for causal inference and has been widely applied to infer tumor susceptibility. MR can effectively demonstrate causal relationships between various factors [5,6,7,8,9]. Expression quantitative trait locus (eQTL) is a type of genetic locus that can affect gene expression levels and can be explored via MR analysis to identify genes causally related to DCIS occurrence.

A comprehensive search of PubMed revealed limited articles on DCIS susceptibility genes. To identify such genes, we integrated the GEO database with MR analysis. This approach not only identified differentially expressed genes (DEGs) but also clarified the causal relationship between genes and DCIS. Furthermore, multivariable analysis of gene expression patterns provided a valuable method for the identification of novel genes involved in DCIS, thereby laying the groundwork for further research and potential clinical applications.

## 2. Materials and Methods

The overall experimental design is illustrated in Figure 1.

### 2.1. GEO Datasets

#### 2.1.1. GEO Dataset Selection

Human GEO datasets for DCIS were downloaded from the GEO database of NCBI (http://www.ncbi.nlm.nih.gov/gds/, accessed on 6 January 2025). All transcriptome datasets were collected and processed for subsequent analysis. A random dataset was selected for validation. Each dataset group contained normal and DCIS subgroups.

#### 2.1.2. GEO Dataset Statistical Analysis

The R software (version 4.4.1) was utilized for data processing. First, the data were normalized and the batch effects were removed using the SVA package, and data visualization was conducted using the PCA package (Appendix A). The limma package was used to screen for DEGs, applying the following criteria: |log(fold change)| > 0.585 and *p*-value < 0.05.

### 2.2. eQTL Dataset Processing

#### 2.2.1. eQTL Dataset Acquisition

The eQTL datasets were obtained from the IEU openGWAS project (https://gwas.mrcieu.ac.uk/, accessed on 7 January 2025). A total of 19,943 eQTL gene-related GWAS datasets were included in this study.

#### 2.2.2. eQTL Statistical Analysis

The TwoSampleMR package was used to process eQTL data. First, linkage disequilibrium (LD) and weak instrumental variable (WIV, F > 10) were excluded. Subsequently, the DCIS-related GWAS dataset “ukb-d-D05” was designated as the outcome variable. Next, four MR methods—MR egger [MRE], weighted median [WM], inverse variance weighted [IVW], and simple mode [SM]—were applied to confirm SNPs of genes associated with the outcome variable. DEGs were screened based on the criteria: *p* ≤ 0.05, consistent direction of odds ratio (OR) values, and absence of horizontal pleiotropy in gene SNPs.

### 2.3. GEO and eQTL Joint Analysis

#### 2.3.1. Acquisition and Validation of Intersection Genes

The VennDiagram package was used to identify overlapping genes, as described in the two aforementioned methods. The TwoSampleMR package was utilized to exclude heterogeneity and pleiotropy of DEGs, generate forest plots and scatter plots, and perform a leave-one-out sensitivity test for each SNP.

#### 2.3.2. GO and KEGG Pathway Analyses

Gene ontology (GO) and Kyoto Encyclopedia of Genes and Genomes (KEGG; http://www.kegg.jp/ or http://www.genome.jp/kegg/, accessed on 10 January 2025) are two widely used methods for studying gene function. Enrichment analysis of DEGs was performed using the enrichplot and clusterProfiler packages, and results were visualized using the ggplot2 package.

#### 2.3.3. Gene Set Enrichment Analysis (GSEA)

Gene set databases (“c2.cp.kegg.Hs.symbols.gmt” and “c5.go.symbols.gmt”) were downloaded from Gene Set Enrichment Analysis (GSEA; https://www.gsea-msigdb.org/gsea/msigdb/collections.jsp, accessed on 7 January 2025). The DEG expression data from GEO datasets were stratified into two groups based on gene expression levels (upper group and lower group). Genes in the DCIS group were retained and those with an expression value of 0 were excluded. For DEGs with two or more expression values, the mean was calculated. Pathway enrichment was considered significant under the following criteria: normalized enrichment score (|NES| > 1), nominal (NOM) *p*-value < 0.05, and FDR-adjusted *q*-value < 0.25. The top five significantly enriched signaling pathways were presented.

#### 2.3.4. Immune Cell Infiltration

Newman AM et al. developed Cell-type Identification by Estimating Relative Subsets of RNA Transcripts (CIBERSORT), a tool enabling the quantification of immune cell abundances [10]. CIBERSORT was used to assess immune cell infiltration in both normal breast and DCIS groups. The immune cell gene abundance file was downloaded from CIBERSORT (https://cibersortx.stanford.edu/, accessed on 12 January 2025), and the results were visualized. The linkET package was applied to illustrate the relationship between DEGs and immune cells.

### 2.4. Data Validation

The study outcomes were validated using another randomly selected GEO dataset containing both normal and DCIS groups. The limma package was used to visualize DEG expression.

### 2.5. Gene Expression Validation

The Human Protein Atlas (HPA; https://www.proteinatlas.org/, accessed on 10 March 2025) is a comprehensive database that displays immunohistochemical results of various tissues and pathological conditions. The accuracy of our findings was verified by examining the expression levels of DEGs in DCIS through this database.

### 2.6. Functional Experimental Validation

#### 2.6.1. Cell Culture

Primary DCIS cells were obtained from fresh postoperative tissue specimens of DCIS patients who provided informed consent, and the study was approved by the Ethics Committee of the First Affiliated Hospital of Fujian Medical University (Approval code: MRCTA, ECFAH of FMU [2021] 368, Approval date: 27 September 2021). Tumor tissues were minced with ophthalmic scissors and then digested with collagenase type IV (1 mg/mL; Sigma-Aldrich, St. Louis, MO, USA) and hyaluronidase (100 U/mL; STEMCELL Technologies, Vancouver, BC, Canada) at 37 °C for 2 h. The cell suspension was filtered through a 100 μm strainer. Cells were seeded in DMEM/F12 medium (Gibco, Grand Island, NY, USA) supplemented with 10% fetal bovine serum (FBS; Gibco, Grand Island, NY, USA), 5 μg/mL insulin (Sigma-Aldrich, St. Louis, MO, USA), 10 ng/mL epidermal growth factor (EGF; PeproTech, Cranbury, NJ, USA) and 1% penicillin-streptomycin (Gibco, Grand Island, NY, USA). The cells were cultured in a humidified incubator at 37 °C with 5% CO_2_. Medium was replaced every 2–3 days. Cells were passaged when they reached 80–90% confluence.

#### 2.6.2. Cell Transfection

Small interfering RNAs (siRNAs) targeting PTPN12, YTHDC2, and MAPKAPK3 (si-PTPN12, si-YTHDC2, and si-MAPKAPK3), along with a control siRNA (designated as siRNA), were commercially obtained from GeneJikai (Shanghai, China). Overexpression plasmids (GPX3-wt, RASA3-wt, TSPAN4-wt) and an empty vector control (designated as vector) were constructed by Servicebio (Wuhan, China). For transfection, Lipofectamine 2000 reagent (Thermo Fisher Scientific, Waltham, MA, USA) was used according to the manufacturer’s protocol to deliver these constructs into two distinct cell lines. Subsequent experimental assays were performed 48 h post-transfection.

#### 2.6.3. Quantitative Real-Time PCR (qRT-PCR)

Total RNA was extracted from transfected cells using TRIzol reagent (Sigma-Aldrich, St. Louis, MO, USA) according to the manufacturer’s protocol. Reverse transcription was performed using the First-Strand cDNA Synthesis Kit (Yeasen, Shanghai, China). qRT-qPCR was conducted with SYBR Green Master Mix (Yeasen, China) on an Applied Biosystems ViiA™7 Real-Time PCR System (Applied Biosystems, Foster City, CA, USA), with three technical replicates per sample. GAPDH served as the endogenous control, and relative gene expression levels were calculated using the 2^−ΔΔCT^ method. The primer sequences used for mRNA detection are listed as follows: PTPN12-forward (5′–3′): ACAGAGCTGCTGAGTCGTCAGAG PTPN12-reverse (5′–3′): ACAGGTGTGGCATTTTCAGGTCC, YTHDC2-forward (5′–3′): CCAGCATTACACCCACCTCAGAAG YTHDC2-reverse (5′–3′): TGGAGGAGAAGGACTAGCACAAGG, MAPKAPK3-forward (5′–3′): GGAAGGTGGTGAGTTGTTCAG MAPKAPK3-reverse (5′–3′): GCCAATATCCCGCATTATCTCTG, GPX3-forward (5′–3′): GAGCTTGCACCATTCGGTCT GPX3-reverse (5′–3′): GGGTAGGAAGGATCTCTGAGTTC, RASA3-forward (5′–3′): ATAGATGGGGAGATTGAGGTT RASA3-reverse (5′–3′): ATCTTCAAACCAAACCCAAAAACTCAATAA, TSPAN4-forward (5′–3′): GCTGTGGCGTCTCCAACTAC TSPAN4-reverse (5′–3′): CTTGGCAGTACATGGTCATGG, GAPDH-forward (5′–3′): AATGGACAACTGGTCGTGGAC GAPDH-reverse (5′–3′): CCC TCCAGGGGATCTGTTTG.

#### 2.6.4. Transwewll Invation Assay

Two primary DCIS cell lines (DCIS-1 and DCIS-2) at passages 3 to 5 were selected and seeded in serum-free medium into the upper chamber of Matrigel-coated Transwell inserts (Millipore, Billerica, MA, USA), while the lower chamber contained medium supplemented with 10% FBS. After 48 h, the cells that had invaded to the lower surface were fixed with methanol, stained with crystal violet, and counted in five randomly selected fields.

### 2.7. Statistics

Statistical analyses were performed using GraphPad Prism 8.0 (GraphPad Software, San Diego, CA, USA). Quantitative data were presented as mean ± standard deviation. Group comparisons were made using Student’s *t*-test (for two groups) or one-way ANOVA (for multiple groups), with a significance threshold set at *p* < 0.05.

## 3. Results

### 3.1. Outcomes of the GEO and eQTL Datasets

Four transcriptome datasets (GSE7882, GSE16873, GSE59246, and GSE21422) were retrieved from GEO, each containing data on both normal and DCIS patients (Appendix A summarizes the GEO database sources, whereas Appendix A contains patient information). DEGs were identified from heatmaps and volcano plots (Figure 2A,B). A total of 2040 DEGs were filtered from the GEO dataset. Subsequently, 19,943 eQTL gene expression data were obtained from IEU openGWAS as exposure factors (Appendix A). The ‘ukb-d-D05’ dataset was used as the outcome variable (Appendix A). After excluding LD and WIV, 26,152 gene-related SNPs were included in subsequent analyses (Appendix A). The MR study was conducted to clarify causal relationships between outcome and exposure data (Appendix A), and the MR results of exposure SNPs are shown in Appendix A. By excluding SNP heterogeneity and pleiotropy (Appendix A), 349 DEGs and 1745 associated SNPs meeting the criteria were finally obtained (Appendix A). Next, the two DEG groups were divided into upregulated and downregulated subtypes, and overlapping genes were selected: four DEGs (LGALS8, PTPN12, YTHDC2, and RNGTT) in the upregulated group and nine (CYB5R2, KLHDC4, APOBEC3G, GPX3, RASA3, TSPAN4, MAPKAPK3, ZFP37 and RAB3IL1) in the downregulated group (Figure 2C,D). MR analyses were performed on SNPs of each DEG, and scatter plots, forest plots, funnel plots, and leave-one-out sensitivity tests were conducted (Appendix A). The forest plot of the overall gene set is shown in Figure 3A, and the chromosomal position of each gene was also visualized (Figure 3B).

### 3.2. GO and KEGG

Thirteen DEGs were enriched in 104 GO terms, including 71 biological process (BP) terms, 5 cellular component (CC) terms, and 28 molecular function (MF) terms (GO enrichment results are shown in Figure 3). Within BP terms, DEGs were enriched in the viral genome replication. For CC terms, DEGs were primarily enriched in ribonucleoprotein granules, focal adhesions, and cell−substrate junctions. In terms of MF terms, DEGs were mainly enriched in integrin binding and phosphoprotein phosphatase activity. KEGG pathway analysis revealed enrichment in amino sugar and nucleotide sugar metabolism (Appendix A).

### 3.3. GSEA

GO and KEGG visualization analyses of 13 DEGs using the aforementioned annotation files and significantly enriched pathways are shown in Figure 4. Based on our results, most DEGs were enriched in immune-related pathways, such as LGALS8, PTPN12, YTHDC2, RNGTT, CYB5R2, KLHDC4, APOBEC3G, GPX3, MAPKAPK3, RASA3 and RAB3IL1. For the upregulated genes, all four genes were associated with the graft versus host disease signaling pathway. Additionally, PTPN12, YTHDC2 and RNGTT were related to antigen processing and presentation signaling pathway; PTPN12 was related to the notch signaling pathway; LGALS8 could act on the JAK-STAT signaling pathway (Figure 5A–D and Appendix A). For the downregulated genes, CYB5R2, KLHDC4, and GPX3 were related to antigen processing and presentation signaling pathway; APOBEC3G could participate in the Toll-like receptor signaling pathway and gamma delta T cell differentiation signaling pathway; MAPKAPK3 could act on the JAK-STAT signaling pathway and the Notch signaling pathway; RAB3IL1 could positively regulate the TOR and TORC1 signaling pathways, and also affect the MAPK signaling pathway (Figure 6A–E and Appendix A).

### 3.4. CIBERSORT Analysis

Based on the GSEA results, we further investigated immune-related cells. Figure 7A,B show the proportions of immune cells in DCIS and normal breast tissue. The results indicated that the activation status of dendritic cell was significantly different between groups. Finally, we explored the relationship between genes and the abundance of other immune cells (Figure 7C).

### 3.5. Statistical Validation

We further validated the results using the GEO dataset (GSE21422), which included data from 19 patients. We randomly selected five individual DCIS samples and five individual normal breast tissue samples to examine the differential expression levels of DEGs (Boxplots are shown in Figure 8). In the validation cohort, significant differences were observed in six genes: three upregulated genes (LGALS8, YTHDC2, and MAPKAPK3) and three downregulated genes (GPX3, RASA3, and TSPAN4).

### 3.6. Validation via the Human Protein Atlas

Using the Human Protein Atlas, we further validated the expression of DEGs in pathological samples. The expression patterns of 8 DEGs were consistent with our predicted results as shown in Figure 9 and Figure 10: 3 upregulated genes (LGALS8, PTPN12, YTHDC2) and 5 downregulated genes (APOBEC3G, CYB5R2, GPX3, KLHDC4, RAB3IL1).

### 3.7. Results of Cellular Experiments

qRT-PCR results confirmed successful transfection (Figure 11B). Transwell invasion assays showed that knockdown of PTPN12, YTHDC2, and MAPKAPK3, as well as overexpression of GPX3, RASA3, and TSPAN4, significantly inhibited the invasive capacity of DCIS cells (Figure 11A,C,D). These experimental findings were consistent with the predicted oncogenic or tumor-suppressive roles of these DEGs identified in the bioinformatics analysis.

## 4. Discussion

Despite being recognized as a precursor to IDC, DCIS has received relatively limited research attention, and its treatment strategies remain ambiguous [1]. Furthermore, it is a significant challenge to distinguish pure DCIS lesion from a related atypical lesion and addressing the substantial intralesional and nuclear grading heterogeneity [11]. Even with the rapid development of breast biopsy techniques and imaging omics, the accuracy of preoperative pathology and diagnostic imaging in identifying underestimated IDC components within DCIS lesions remains suboptimal. Consequently, some patients who undergo breast-conserving surgery may experience ipsilateral IDC recurrence within a few years [12,13,14]. Therefore, elucidating the molecular mechanisms underlying DCIS progression is crucial not only for identifying high-risk DCIS lesions but also for tracing the origins of this condition [15]. Although progress has been made in understanding its molecular mechanism, the causal links to subsequent diseases remain undetermined, necessitating further exploration to strengthen the association between DCIS and IDC.

With the expansion of the GWAS database, MR has emerged as a reliable approach for inferring causal relationships between exposure factors and outcome indicators. Previous meta-analyses have paved the way for genetic-level investigations into the origins of DCIS [16,17]. In this study, we identified 13 DEGs by integrating eQTL and GEO datasets and predicted their functions Via GO and KEGG enrichment analyses. We found that these DEGs were mainly enriched in pathways related to the cell cycle, proliferation, and immune processes (e.g., antigen processing and presentation, suggesting a pivotal role of the immune microenvironment in early-stage DCIS).

We subsequently focused on the immune functions of these DEGs. Prior studies have linked several genes (LGALS8, PTPN12, YTHDC2, APOBEC3G, GPX3, RASA3, and TSPAN4) to immune responses, highlighting the impact of the tumor microenvironment (TME) on DCIS [18,19,20,21,22,23,24]. Our GSEA analysis demonstrated that APOBEC3G was associated with “Gamma delta T cell differentiation”. Gamma delta T cells can directly recognize and eliminate tumor cells without relying on the stimulation by specific antigens in tumor immunotherapy [25], playing a key role in the tumor progression [26,27]. Dysfunction or exhaustion of gamma delta T cell in the TME may contribute to early immune escape in DCIS [28]. Additionally, “antigen processing and presentation” was another critical theme involving DEGs (CYB5R2, GPX3, KLHDC4, PTPN12, RNGTT, and YTHDC2). Given previous evidence that enhancing antigen presentation improves the efficacy of immunotherapy across various tumors [29,30,31], our results suggest that targeting DEGs involved in this process could be a viable strategy for preventing early breast cancer and augmenting local anti-tumor immunity.

Notably, PTPN12, identified as a tumor suppressor gene (TSG), exhibited upregulation in our DCIS samples—contrasting with its downregulation in invasive breast cancer (IBC)—indicating its potential as a predictive biomarker for DCIS progression [32]. CIBERSORT analysis further revealed significantly reduced dendritic cell activation in DCIS, which may represent a key mechanism underlying early immune evasion. This finding aligns with a recent study showing improved therapeutic outcomes in Her-2-positive early breast cancer patients receiving dendritic cell-targeted vaccines [33]. Moreover, emerging evidence links CYB5R2 to IBC-related ferroptosis, KLHDC4 loss to impaired cell apoptosis [34,35], and MAPKAPK3 to cellular autophagy [36]. However, additional experiments are needed to validate these associations across different breast cancer subtypes.

The interplay of diverse signaling cascades likely contributes to DCIS development. For instance, the toll-like receptor signaling pathway has demonstrated therapeutic potential in colorectal cancer, yet its role in breast cancer remains unclear and warrants further investigation [37]. The same signaling pathway may also be involved in breast cancer, but whether targeting it could yield benefits remains unknown, and necessitating additional research [38]. Similarly, KLHDC4’s association with the WNT pathway—implicated in breast cancer progression, immune modulation, invasion, and metastasis—highlights the complexity of DEG-driven mechanisms [39]. Overall, these findings underscore the multifaceted regulation of DCIS and suggest that identifying specific DEGs may facilitate risk stratification and reduce overtreatment with drugs.

In addition to the preceding bioinformatics predictions, experimental validation was performed through in vitro cellular functional studies to confirm the computational findings. Functional experiments validated the biological roles of the bioinformatics-identified DEGs (PTPN12, YTHDC2, GPX3, RASA3, and TSPAN4) in DCIS progression. Specifically, a previous study demonstrated that overexpression of GPX3 inhibits the progression of breast cancer cells in vitro [40], which is consistent with our study. However, while MAPKAPK3 was downregulated in the DEGs analysis, its expression was upregulated in the validation cohort, and cellular invasion assays suggested its potential involvement in DCIS advancement. The paradoxical result may be associated with tumor complexity, multidimensional gene regulation, and experimental heterogeneity. Therefore, further investigation is warranted to elucidate whether MAPKAPK3 has potential as a DCIS progression marker or therapeutic target.

However, current bulk data-based analyses cannot elucidate how DCIS cells communicate with the TME and the spatial heterogeneity of different cell subpopulations—both of which are crucial for understanding complex carcinogenic processes [41]. Single-cell analysis (SCA) offers irreplaceable advantages in this regard: it can capture gene expression profiles, spatial distribution, and intercellular communication at the single-cell level, thereby revealing subtle differences in molecular mechanisms during DCIS progression and variation in drug responses [42]. For instance, SpaRx has successfully revealed the heterogeneity of drug sensitivity in cells at different l intratumoral locations Via SCA [43], while the DRMref database has systematically compiled single-cell characteristics related to drug resistance in various cancers based on SCA data [44]. These findings demonstrate the timeliness and necessity of SCA in exploring complex disease mechanisms. Therefore, future studies can further verify the expression and function of core DEGs in different DCIS cell subpopulations by integrating SCA, thereby facilitating the translation of research findings into clinical practice.

This study has several limitations. Firstly, the inconsistent differential expression of some DEGs across validation datasets may indicate the existence of uncharacterized DCIS subtypes, which could affect the reliability of the findings. Secondly, the research primarily relies on bioinformatics analyses and basic cellular functional experiments. To strengthen experimental verification and explore the pathogenesis of DCIS more deeply, additional clinical samples and rigorous molecular biology experiments are required. Lastly, we used primary DCIS cells, which retain the original traits of tumors and more accurately recapitulate the invasive potential of DCIS. However, inherent inter-patient heterogeneity may introduce variability, and passage culture in vitro may induce phenotypic drift in tumor cells. Such factors may collectively reduce experimental reproducibility.

## 5. Conclusions

This study identifies potential driver genes and associated pathways in DCIS, laying a foundation for future research on the DCIS-IDC relationship. Our findings highlight the importance of understanding the origins of DCIS and provide genetic-level insights for subtype classification.

## Figures and Tables

**Figure 1 cimb-47-00747-f001:**
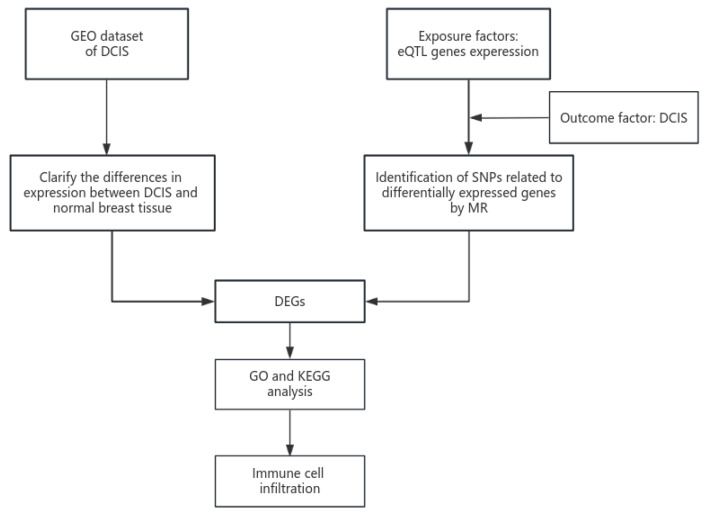
Experimental design of GEO combined with the Mendelian randomization study on differentially expressed genes in DCIS.

**Figure 2 cimb-47-00747-f002:**
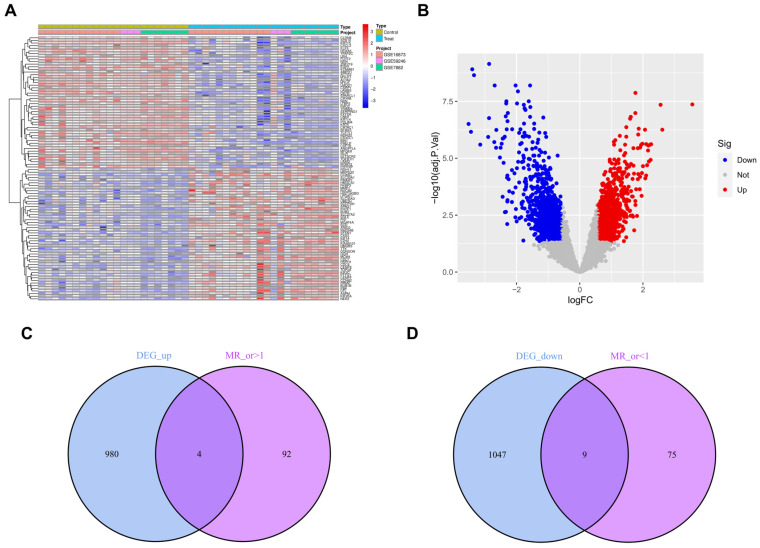
(**A**) Heatmap showing differentially expressed genes (DEGs) between ductal carcinoma in situ (DCIS) and normal breast tissue. (**B**) Plot of DEGs between DCIS and normal breast tissue. (**C**) Number of overlapping upregulated DEGs. (**D**) Number of overlapping downregulated DEGs.

**Figure 3 cimb-47-00747-f003:**
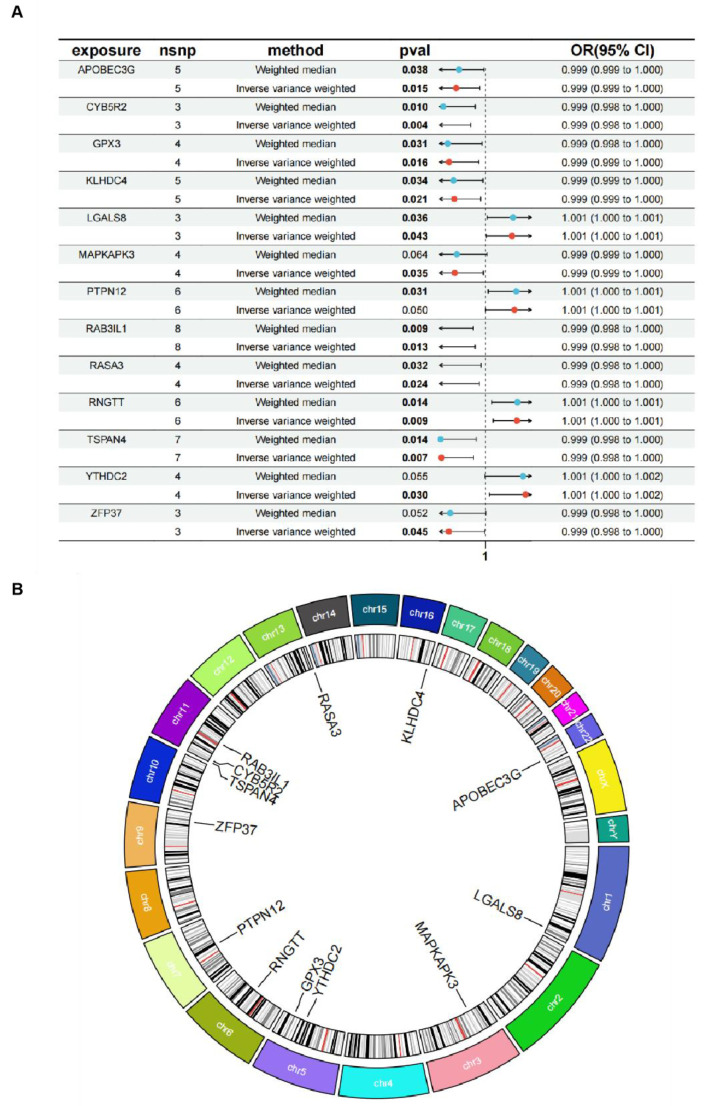
(**A**) Forest plot differentially expressed genes (DEGs) from Mendelian randomization (MR) analysis. (**B**) Chromosomal positions of each DEG.

**Figure 4 cimb-47-00747-f004:**
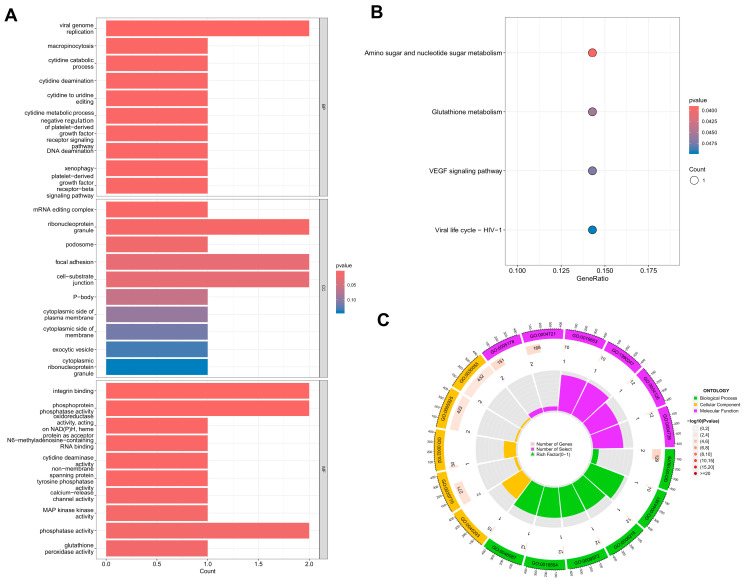
(**A**) Results of gene ontology (GO) enrichment analysis. (**B**) Results of Kyoto Encyclopedia of Genes and Genomes (KEGG) enrichment analysis. (**C**) Visualization of enrichment results and gene counts.

**Figure 5 cimb-47-00747-f005:**
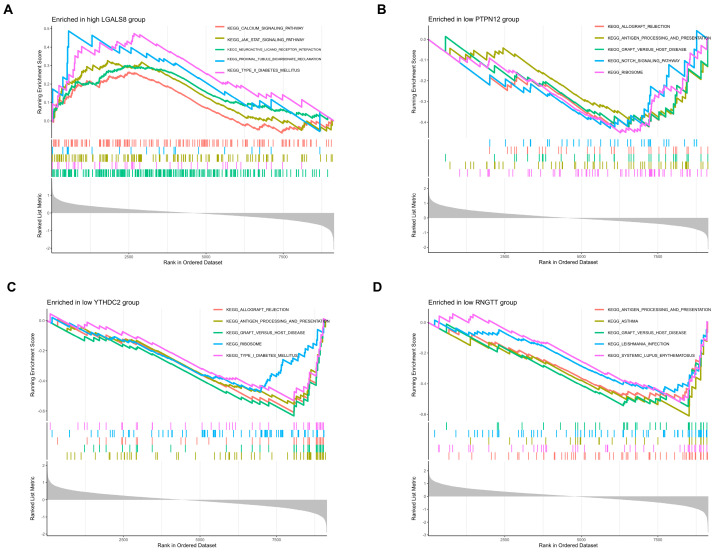
(**A**–**D**) Gene Set Enrichment Analysis (GSEA) results for upregulated differentially expressed genes (DEGs) associated with immune cell infiltration. Additional details are provided in Appendix A.

**Figure 6 cimb-47-00747-f006:**
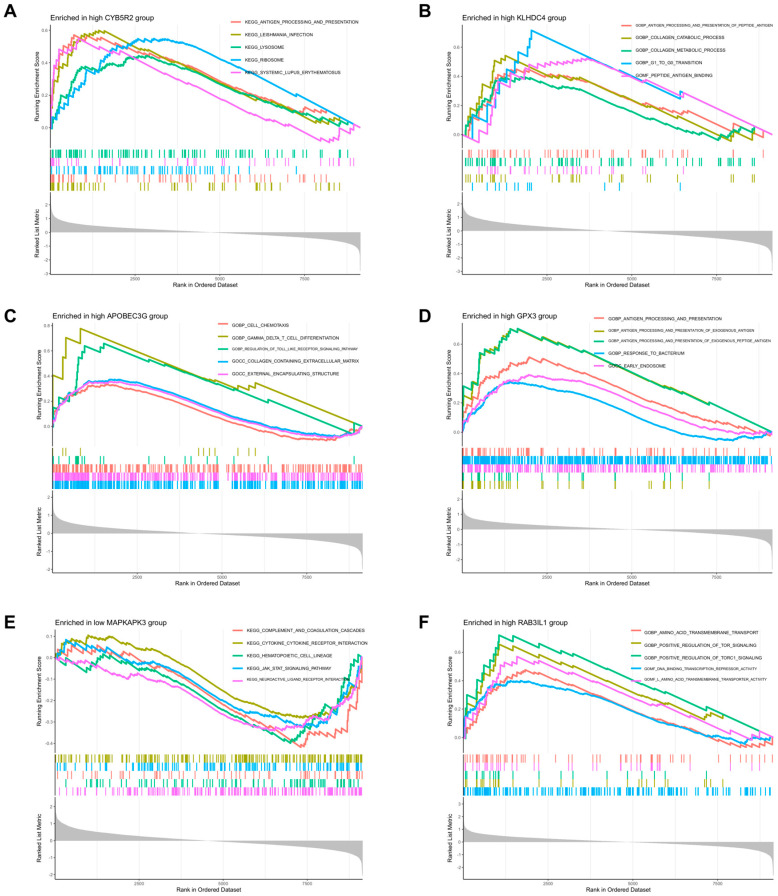
(**A**–**F**) Gene Set Enrichment Analysis (GSEA) results for downregulated differentially expressed genes (DEGs) associated with immune cell infiltration. Additional details are provided in Appendix A.

**Figure 7 cimb-47-00747-f007:**
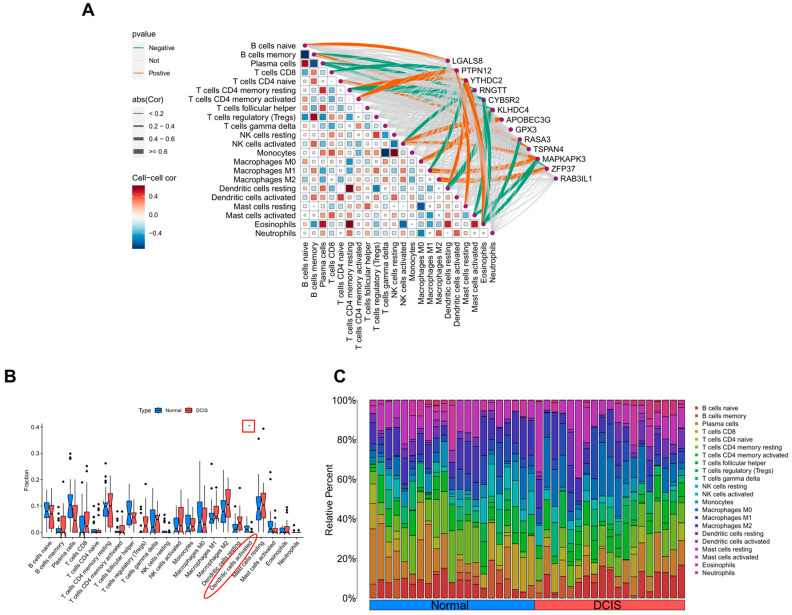
(**A**) Relationship between differentially expressed genes (DEGs) and various immune cell populations (green lines indicate negative regulation, red lines indicate positive regulation; line thickness correlates with the strength of the relationship). (**B**) Differences in immune cell abundance between the DCIS group and normal breast tissue group (* *p* < 0.05). (**C**) Heatmaps showing immune cell infiltration across different groups.

**Figure 8 cimb-47-00747-f008:**
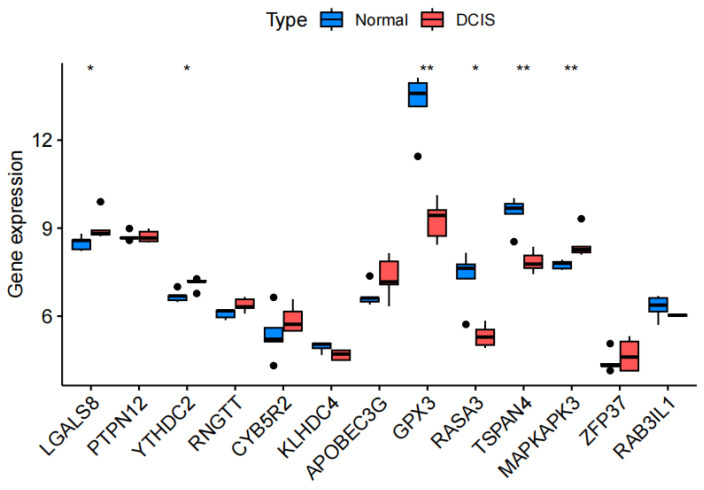
Differential expression of differentially expressed genes (DEGs) between the validation cohorts. Blue for normal tissue, red for DCIS (ductal carcinoma in situ) tissue. * *p* < 0.05, ** *p* < 0.01.

**Figure 9 cimb-47-00747-f009:**
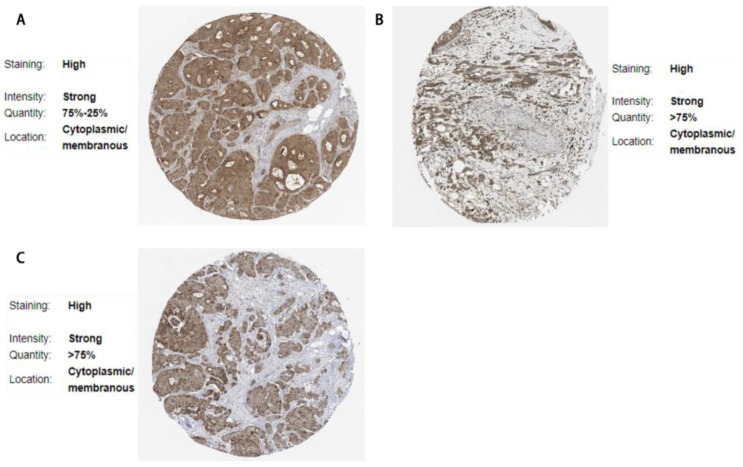
Upregulated differentially expressed genes (DEGs) in the Human Protein Atlas (HPA). (**A**) LGALS8; (**B**) PTPN12; (**C**) YTHDC2.

**Figure 10 cimb-47-00747-f010:**
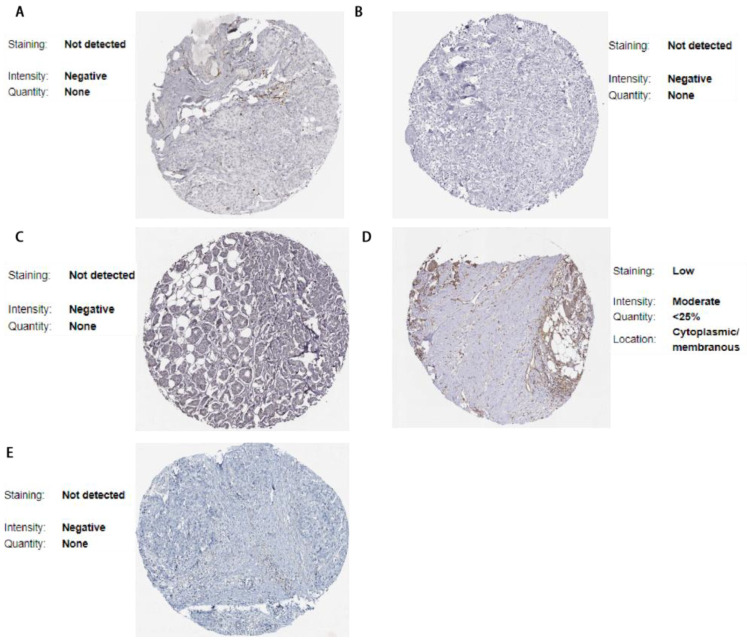
Downregulated differentially expressed genes (DEGs) in the Human Protein Atlas (HPA). (**A**) APOBEC3G; (**B**) CYB5R2; (**C**) GPX3; (**D**) KLHDC4; (**E**) RAB3IL1.

**Figure 11 cimb-47-00747-f011:**
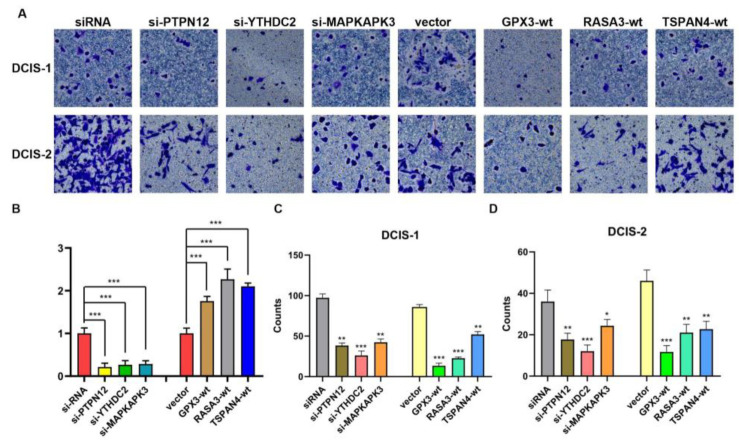
Effects of silencing PTPN12, YTHDC2, and MAPKAPK3 and overexpressing GPX3, RASA3, and TSPAN4 on DCIS cell invasion, determined using Transwell invasion assays. (**A**) Representative images of Transwell assays for DCIS-1 and DCIS-2 cells under different genetic manipulation conditions, scale bar, 50 μm.; (**B**) qRT-PCR results confirming s efficiency and overexpression levels; (**C**,**D**) Quantitative counts of invasive DCIS-1 (**C**) and DCIS-2 (**D**) cells from Transwell assays, respectively. Data are presented as mean ± SD. Statistical significance was assessed by one-way ANOVA; *** *p* < 0.001, ** *p* < 0.01, * *p* < 0.05. siRNA, small interfering RNA; wt, overexpressing type; DCIS, primary ductal carcinoma in situ cell.

## Data Availability

The original contributions presented in the study are included in the article/Appendix A, further inquiries can be directed to the corresponding authors.

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
