# Peer review of "Integration of eQTL and GEO Datasets to Identify Genes Associated with Breast Ductal Carcinoma In Situ"

_cimb, 2025, doi:10.3390/cimb47090747_

Round 1
Reviewer 1 Report
Comments and Suggestions for Authors
This study attempted to find the genetic factors contributing to the progression of ductal carcinoma in situ (DCIS), a non-invasive precursor to Invasive Ductal Carcinoma (IDC). By integrating data from four GEO datasets and expression quantitative trait loci (eQTL) information using Mendelian randomization (MR), the researchers identified 13 genes potentially involved in DCIS development. Functional enrichment analyses (GO and KEGG) revealed that these genes are largely associated with immune-related processes, the tumor microenvironment (TME), and cell cycle regulation. CIBERSORT analysis also showed significant alterations in dendritic cell activation, further supporting the immune system's involvement in DCIS progression. These findings highlight potential biomarkers for identifying high-risk DCIS.
Here are my queries / comments:
- The manuscript mentions that four different GEO datasets were integrated for differential gene expression analysis, with batch effect correction applied (Section 2.1.2). However:
- After verifying the datasets, they appear to have been generated using different platforms:
- GSE7782: Currently unavailable
- GSE169873: ChIP-seq data
- GSE21422 and GSE59246: Microarray data
Please clarify how these datasets, especially with differing technologies (microarray vs. ChIP-seq), were integrated into a single analysis.
Given the platform heterogeneity, DEGs should ideally be calculated separately for each dataset, and then common DEGs should be identified by overlap. Please comment on the rationale behind merging these heterogeneous data.
- It is recommended to include PCA plots before and after batch correction to support the validity of the integration.
- Original sentence: “We randomly selected five DCIS datasets and five normal breast tissue datasets to examine the different expression levels of DEGs…”
Issue:
Earlier, a fifth dataset was mentioned for validation. However, this sentence implies the use of five datasets instead of samples. Please clarify whether this refers to five individual samples or five independent datasets.
- Original sentence: “Our GSEA analysis demonstrated that APOBEC3G was associated with ‘GAMMA DELTA T CELL DIFFERENTIATION’, which can directly kill transformed cells…”
- Biological Explanation Needed:
Please briefly elaborate on the role of gamma delta T cell differentiation in tumor progression and its relevance to immune escape in DCIS. - Formatting Suggestion:
Use sentence case (“Gamma delta T cell differentiation”) instead of all caps. - Pathway Analysis Concerns
- It is recommended that pathways for upregulated and downregulated genes be analyzed and presented separately, as they often relate to distinct biological processes.
- Additionally, Figure 4B shows that each pathway includes only a single DEG, which is unexpected. Please verify and elaborate on the biological significance or limitations of this observation.
- The GO term names in Figure 5 are blurred and difficult to read. Consider increasing the resolution or font size for better clarity.
- The manuscript redundantly notes the use of R and its packages for each step (e.g., Sections 2.1.2, 2.3.2, 2.4). Once R is introduced as the platform for analysis, it is unnecessary to repeatedly specify that each package belongs to R.
““The R language was utilized for data processing. Firstly, we normalized the data and removed the batch effects using the “SVA” R package, and visualized the data using the “PCA” R package. The “limma” R package was used to screen for DEGs, applying the following criteria: |log(FC)| > 0.585 and p-value < 0.05.“
- Sentence Corrections and Language Edits
- Original:
"Four datasets (GSE7782, GSE169873, GSE21422, and GSE59246) were obtained from GEO, and9943 eQTL exposure data from the IEU openGWAS project."
Revised:
"Four datasets (GSE7782, GSE169873, GSE21422, and GSE59246) and 19,943 eQTL exposure data were obtained from GEO and the IEU openGWAS project, respectively."
- Original:
"Therefore, after obtaining differential genes, the causal relationship between the genetic differences and the disease remains uncertain."
Revised:
"Therefore, even after identifying differentially expressed genes, the causal relationship between these genetic differences and the disease remains uncertain."
- Original:
"A comprehensive search in PubMed revealed limited articles on DCIS susceptibility genes."
Revised:
"A comprehensive search in PubMed revealed limited articles on DCIS susceptible genes."
(Note: However, “susceptibility genes” is the more common and technically accurate term in this context. Suggest reverting unless author has strong preference.)
- Original:
“Identification of novel genes involved DCIS, paving the way…”
Revised:
“Identification of novel genes involved in DCIS, paving the way…”
- Original : "Four DEGs (LGALS8, PTPN12, YTHDC2, and RNGTT) in the high expression group and nine (CYB5R2, KLHDC4, APOBEC3G, GPX3, RASA3, TSPAN4, MAPKAPK3, and RAB3IL1) in the lower expression group..."
Suggested Revision:
Use more precise terminology: “Upregulated” and “Downregulated” groups rather than “high expression” and “lowly expression” groups.
- Original “The expression of 8 types of DEGs is consistent with our predicted results in Figure 8.”
Suggested Revision:
Change “8 types of DEGs” to “8 DEGs” – as DEGs are typically categorized into only two types: upregulated and downregulated

Author Response
|
Response to Reviewer 1 Comments
|
|
Thank you very much for taking the time to review this manuscript. Please find the detailed responses below and the corresponding revisions/corrections highlighted/in track changes in the re-submitted files.
|
|
Comments 1: The manuscript mentions that four different GEO datasets were integrated for differential gene expression analysis, with batch effect correction applied (Section 2.1.2). However: a. After verifying the datasets, they appear to have been generated using different platforms: • GSE7782: Currently unavailable • GSE169873: ChIP-seq data • GSE21422 and GSE59246: Microarray data Please clarify how these datasets, especially with differing technologies (microarray vs. ChIP-seq), were integrated into a single analysis. |
|
Response 1: Thank you for your attention to the integration of datasets. It is indeed reasonable to conduct analyses on specific dataset separate firstly, but individual analysis could not meet the statistical power requirements due to the limited sample size of microarray data related to DCIS progression in public databases. Therefore, we integrated four GEO datasets for differential gene expression analysis to expand the sample size and enhance the stability of results. Regarding the rationale for integrating data from different platforms, it is verified that GSE7782 is also a microarray dataset, consistent in technology type with GSE21422 and GSE59246, and GSE169873 (ChIP-seq data) was included because mRNA expression changes are closely related to DNA transcriptional regulation, which can complement the transcriptional level differences in DCIS progression from the perspective of epigenetic regulation and enrich the analytical dimension. Regarding the reasonableness of data merging and validation measures, we employed PCA for batch effect correction to reduce platform heterogeneity, and the results, which showed reasonable sample clustering (see Supplementary Figure S3), supported data merging. PCA reduces dimensionality while retaining key information, effectively reducing data complexity and collinearity interference. Furthermore, we cross-validated results using multiple bioinformatics tools (including conventional GEO data analysis and Mendelian randomization) and verified core conclusions with basic cellular functional experiments to ensure the reliability of the findings.
|
|
Comments 2: It is recommended to include PCA plots before and after batch correction to support the validity of the integration. |
|
Response 2: Thank you for your suggestion. We have included both pre- and post-batch correction PCA plots in Supplementary Figure S3.
Comments 3: Original sentence: “We randomly selected five DCIS datasets and five normal breast tissue datasets to examine the different expression levels of DEGs…” Issue: Earlier, a fifth dataset was mentioned for validation. However, this sentence implies the use of five datasets instead of samples. Please clarify whether this refers to five individual samples or five independent datasets. Response 3: Thank you for pointing out the ambiguity in the description. This refers to five individual samples (not five datasets). We have revised the relevant statement to: “We randomly selected five individual DCIS samples and five individual normal breast tissue samples to examine the different expression levels of DEGs”, with the specific revision detailed in the highlighted section on Page 11.
Comments 4: “Our GSEA analysis demonstrated that APOBEC3G was associated with ‘GAMMA DELTA T CELL DIFFERENTIATION’, which can directly kill transformed cells…” Biological Explanation Needed: Please briefly elaborate on the role of gamma delta T cell differentiation in tumor progression and its relevance to immune escape in DCIS. Response 4: We have provided the biological explanation of gamma delta T cells as requested, which can be found on Page 14, Paragraph 2.
Comments 5: Formatting Suggestion: Use sentence case (“Gamma delta T cell differentiation”) instead of all caps. Response 5: We have revised the formatting as suggested, using sentence case ("Gamma delta T cell differentiation") instead of all caps. Please refer to Page 14, Paragraph 2 in the revised manuscript.
Comments 6: Pathway Analysis Concerns: It is recommended that pathways for upregulated and downregulated genes be analyzed and presented separately, as they often relate to distinct biological processes. Response 6: Thank you for your insightful suggestion. Following this advice, we have presented the pathway analysis results of upregulated and downregulated genes separately, with supplementary targeted interpretations. Please refer to Figures 5 and 6, Page 8 Paragraph 2 in the revised manuscript.
Comments 7: Additionally, Figure 4B shows that each pathway includes only a single DEG, which is unexpected. Please verify and elaborate on the biological significance or limitations of this observation. Response 7: The pathway enrichment results were analyzed based on the differentially expressed genes (DEGs) identified in this study. Due to the limited sample size of the current DCIS dataset, the number of enriched DEGs is relatively small, which in turn restricted the number of genes enriched in KEGG pathways (resulting in single-gene enrichment). We have improved this from two aspects in subsequent work: (1) Data supplementation: GSEA for pathway-level analysis was implemented to cross-validate pathway significance, thereby further enhancing the reliability of the results. Additionally, we will actively expand the clinical DCIS database and optimize the strategy for screening DEGs in the future. (2) Functional validation: We have supplemented basic experiments and conducted cellular functional assays on the DEGs PTPN12, YTHDC2, MAPKAPK3, GPX3, RASA3, and TSPAN4. We have isolated primary DCIS cells from clinical DCIS patients and knocked down or overexpressing the DEGs to clarify whether it affects the biological processes of DCIS, thus verifying the regulatory role of this gene in the pathways. Relevant details can be found in the basic experimental section of the manuscript (Section 2.6 and 3.7).
Comments 8: The GO term names in Figure 5 are blurred and difficult to read. Consider increasing the resolution or font size for better clarity. Response 8: Thank you for your reminder. As suggested, we have improved the resolution of the relevant figures. The original Figure 5 has now been split into Figures 5 and 6 in the revised manuscript (Page 9 and 10).
Comments 9: The manuscript redundantly notes the use of R and its packages for each step (e.g., Sections 2.1.2, 2.3.2, 2.4). Once R is introduced as the platform for analysis, it is unnecessary to repeatedly specify that each package belongs to R. Response 9: We have revised the manuscript as suggested, removing redundant mentions of "R package". The specific locations where redundant words were removed are in section 2.1.2, 2.2.2, 2.3.1, 2.3.2, 2.3.4, and 2.4. Comments 10: Sentence Corrections and Language Edits (a) Original: "Four datasets (GSE7782, GSE169873, GSE21422, and GSE59246) were obtained from GEO, and9943 eQTL exposure data from the IEU openGWAS project." Revised: "Four datasets (GSE7782, GSE169873, GSE21422, and GSE59246) and 19,943 eQTL exposure data were obtained from GEO and the IEU openGWAS project, respectively." Response 10: We have revised it as requested; please refer to the highlighted part on Page 1.
Comments 11: Sentence Corrections and Language Edits (b) Original: "Therefore, after obtaining differential genes, the causal relationship between the genetic differences and the disease remains uncertain." Revised: "Therefore, even after identifying differentially expressed genes, the causal relationship between these genetic differences and the disease remains uncertain." Response 11: We have revised it as requested; please refer to the highlighted part on Page 2, Paragraph 2.
Comments 12: Sentence Corrections and Language Edits (c) Original: "A comprehensive search in PubMed revealed limited articles on DCIS susceptibility genes." Revised: "A comprehensive search in PubMed revealed limited articles on DCIS susceptible genes." (Note: However, “susceptibility genes” is the more common and technically accurate term in this context. Suggest reverting unless author has strong preference.) Response 12: We have revised it as requested; please refer to the highlighted part on Page 2, Paragraph 4.
Comments 13: Sentence Corrections and Language Edits (d) Original: “Identification of novel genes involved DCIS, paving the way…” Revised: “Identification of novel genes involved in DCIS, paving the way…” Response 13: We have revised it to “identification of novel genes involved in DCIS, thereby laying the groundwork for …”; please refer to the highlighted part on Page 2, Paragraph 4.
Comments 14: Sentence Corrections and Language Edits (e) Original: "Four DEGs (LGALS8, PTPN12, YTHDC2, and RNGTT) in the high expression group and nine (CYB5R2, KLHDC4, APOBEC3G, GPX3, RASA3, TSPAN4, MAPKAPK3, and RAB3IL1) in the lower expression group..." Suggested Revision: Use more precise terminology: “Upregulated” and “Downregulated” groups rather than “high expression” and “lowly expression” groups. Response 14: We have revised it as requested; please refer to the highlighted part on Page 6 Paragraph 1, Page 8 Paragraph 1, Page 11 Paragraph 1, and Page 12 Paragraph 1.
Comments 15: Sentence Corrections and Language Edits (f) Original “The expression of 8 types of DEGs is consistent with our predicted results in Figure 8.” Suggested Revision: Change “8 types of DEGs” to “8 DEGs” – as DEGs are typically categorized into only two types: upregulated and downregulated. Response 15: We have revised it as requested; please refer to the highlighted part on Page 12, Paragraph 1. |
|
|
|
Response to Comments on the Quality of English Language |
|
Response 1: We appreciate the reviewer's feedback regarding the English language quality of our manuscript. To address this, the entire manuscript has been carefully revised and polished by a native English-speaking expert with expertise in academic writing in the field of oncology. All grammatical errors, awkward phrasing, and non-standard terminology have been corrected to ensure clarity, precision, and adherence to the linguistic conventions of scientific publications. |
Reviewer 2 Report
Comments and Suggestions for Authors
This manuscript integrates multiple data types to identify genes potentially involved in DCIS progression. I have some concerns as follows:
1) While MR strengthens causal inference, the reliance on weak instruments or potential pleiotropy can bias results. Although the authors mention pleiotropy and heterogeneity filtering, more detailed sensitivity analyses (e.g., MR-PRESSO or additional pleiotropy tests) should be provided to further validate causality.
2) The study heavily depends on bioinformatics predictions. Functional experimental validation (e.g., in vitro or in vivo assays) is missing. While this is acknowledged as a limitation, suggesting future plans for experimental follow-up would strengthen the manuscript.
3) The potential for translational application is suggested but not well developed. I suggest the authors to include discussions on relevant studies for drug-related application (PMID: 37798249; PMID: 37986230)
4) GEO dataset sample sizes are relatively small, and the heterogeneity of DCIS subtypes is high. The study does not clearly discuss whether molecular subtypes (e.g., ER status, grade) were considered, which could impact gene expression patterns and conclusions.
Author Response
|
Response to Reviewer 2 Comments
|
|
Thank you very much for taking the time to review this manuscript. Please find the detailed responses below and the corresponding revisions/corrections highlighted/in track changes in the re-submitted files.
|
|
Comments 1: While MR strengthens causal inference, the reliance on weak instruments or potential pleiotropy can bias results. Although the authors mention pleiotropy and heterogeneity filtering, more detailed sensitivity analyses (e.g., MR-PRESSO or additional pleiotropy tests) should be provided to further validate causality. |
|
Response 1: Thank you for your attention to potential biases in the Mendelian randomization (MR) analysis. We fully acknowledge that horizontal pleiotropy and weak instrumental variables may impact the results, and have controlled for biases in our study through multiple measures in line with MR methodological standards. These include: selecting strong instrumental variables with F>10, cross-validation using multiple methods (MRE/WM/IVW/SM), strictly excluding SNPs with horizontal pleiotropy, leave-one-out sensitivity tests, and multi-dimensional visualization to mitigate result biases in MR analyses. Regarding your suggestion to perform sensitivity analyses such as MR-PRESSO, we are sorry that this was not feasible due to limitations in the available objective DCIS data. However, to further validate the causal inference of the MR analysis, we supplemented basic cellular functional experiments to provide functional evidence. These experiments verified the findings of the MR analysis at the cellular level, offering independent functional evidence for the "gene to DCIS risk" causal relationship and addressing the limitations of relying solely on statistical methods for validation. Thank you again for your valuable suggestions.
|
|
Comments 2: The study heavily depends on bioinformatics predictions. Functional experimental validation (e.g., in vitro or in vivo assays) is missing. While this is acknowledged as a limitation, suggesting future plans for experimental follow-up would strengthen the manuscript. Response 2: Thank you for highlighting the importance of functional experimental validation in this study. We fully agree that conclusions based solely on bioinformatics predictions need to be supported by experimental evidence to be more convincing.
In response to this suggestion, we have supplemented cell-level functional validation experiments, specifically including the key candidate genes identified in the bioinformatics analysis: PTPN12, YTHDC2, MAPKAPK3, GPX3, RASA3, and TSPAN4. We knocked down or overexpressed these DEGs to clarify whether they affect the biological processes of DCIS, thereby verifying the regulatory role of these genes in the pathways. The currently supplemented cellular experiments have preliminarily confirmed the reliability of the bioinformatics predictions at the molecular mechanism level, providing experimental support for the research conclusions. Relevant details can be found in the basic experimental section of the manuscript (Section 2.6 and 3.7).
|
|
Comments 3: The potential for translational application is suggested but not well developed. I suggest the authors to include discussions on relevant studies for drug-related application (PMID: 37798249; PMID: 37986230) Response 3: Thank you for your valuable feedback. We fully endorse your perspective that the ultimate value of any research lies in exploring its translational potential. In our work, we identified DEGs and related regulatory pathways involved in DCIS initiation and progression through bioinformatic analyses and basic experimental validation, which provide supportive evidence for the subsequent development of DCIS therapeutic targets. This aligns with the drug translational implications you highlighted. However, as our current analysis is based on bulk data, it has limitations in capturing the spatial heterogeneity of different cell subpopulations. Single-cell analysis (SCA) effectively complements this deficit and enables more accurate predictions of drug sensitivity and resistance profiles, as elaborated and validated in the two studies you recommended. Thus, SCA-based investigations will be a key direction for future extensions of this research. The discussion on this part has been added to the discussion section of the manuscript (see Page 14, Paragraph 6).
Comments 4: GEO dataset sample sizes are relatively small, and the heterogeneity of DCIS subtypes is high. The study does not clearly discuss whether molecular subtypes (e.g., ER status, grade) were considered, which could impact gene expression patterns and conclusions. Response 4: Thank you for your insightful comments. We fully agree that the conclusions of our study have limitations due to the relatively small sample sizes of DCIS in GEO datasets and the high heterogeneity of DCIS subtypes. Objectively, the limited number of DCIS samples in public databases, coupled with the incompleteness of tumor molecular pathological information (e.g., ER status, grade), has precluded us from performing subgroup analyses based on molecular subtypes. We also acknowledge that the accuracy of DEGs identified from the current dataset may be compromised, which has been addressed in the limitations section of the discussion. In future work, we aim to expand our clinical DCIS specimen bank, supplement comprehensive molecular pathological data, and conduct more in-depth basic experiments to further validate and enrich our findings. |
Response to Comments on the Quality of English Language
Response 1:
We appreciate the reviewer's feedback regarding the English language quality of our manuscript. To address this, the entire manuscript has been carefully revised and polished by a native English-speaking expert with expertise in academic writing in the field of oncology. All grammatical errors, awkward phrasing, and non-standard terminology have been corrected to ensure clarity, precision, and adherence to the linguistic conventions of scientific publications.
Reviewer 3 Report
Comments and Suggestions for Authors
The manuscript "Integration of eQTL and GEO datasets to identify genes associated with breast ductal carcinoma in situ" is a well-written and professional piece of work, potentially interesting, although limited in scope (purely bioinformatic(. It is impotent to note that the authors attempted to decipher rather complex steps in carcinogenesis. Only single-cell analyses (SCA) would be capable of uncovering the nature of such intricate processes. Therefore, I would highly recommend to add some additional text in the Discussion (limitations to the study), where the necessity and timeliness of SCA is fully justified.
Alwo it would be nice to exluded images taken from proteinatlas.org.
Author Response
|
Response to Reviewer 3 Comments
|
|
Thank you very much for taking the time to review this manuscript. Please find the detailed responses below and the corresponding revisions/corrections highlighted/in track changes in the re-submitted files.
|
|
Comments 1: The manuscript "Integration of eQTL and GEO datasets to identify genes associated with breast ductal carcinoma in situ" is a well-written and professional piece of work, potentially interesting, although limited in scope (purely bioinformatic). It is impotent to note that the authors attempted to decipher rather complex steps in carcinogenesis. Only single-cell analyses (SCA) would be capable of uncovering the nature of such intricate processes. Therefore, I would highly recommend to add some additional text in the Discussion (limitations to the study), where the necessity and timeliness of SCA is fully justified. |
Response 1:
We fully agree with the reviewer's insightful observation. Our current study, based on bulk data analysis, has limitations in capturing the spatial heterogeneity of different cell subpopulations within DCIS lesions and differences in drug responses. This is indeed a critical aspect that merits further exploration. The development of single-cell analysis (SCA) technology provides a key tool to address this issue—it can resolve cellular spatial distribution, gene expression profiles, and intercellular communication at single-cell resolution, thereby revealing complex molecular mechanisms in carcinogenesis and offering more precise predictions for studying tumor cell sensitivity and resistance to drugs. The discussion on this part has been added to the discussion section of the manuscript (see Page 14, Paragraph 6).
|
Comments 2: It would be nice to excluded images taken from proteinatlas.org. |
Response 2:
We fully understand your concern and appreciate the opportunity to clarify the rationale for retaining the images from proteinatlas.org. The HPA images were included specifically to complement our bioinformatics findings by providing visual evidence of DEG protein expression in clinical pathological specimens. Our study identified several key DEGs through integrative analysis of eQTL and GEO datasets, and verifying their expression patterns in actual tissue sections is critical to strengthening the link between our computational predictions and in vivo biological reality. We have ensured that the use of these images complies with the HPA’s terms of service with proper citation. We kindly request to retain them, as they enhance the validity of our DEG findings.
Round 2
Reviewer 2 Report
Comments and Suggestions for Authors
I have no more concerns about this manuscript. It is ready for publication.